# Risk Stratification for Management of Solitary Fibrous Tumor/Hemangiopericytoma of the Central Nervous System

**DOI:** 10.3390/cancers15030876

**Published:** 2023-01-31

**Authors:** Connor J. Kinslow, Ali I. Rae, Prashanth Kumar, Guy M. McKhann, Michael B. Sisti, Jeffrey N. Bruce, James B. Yu, Simon K. Cheng, Tony J. C. Wang

**Affiliations:** 1Department of Radiation Oncology, Vagelos College of Physicians and Surgeons, Columbia University Irving Medical Center, 622 West 168th Street, BNH B011, New York, NY 10032, USA; 2Herbert Irving Comprehensive Cancer Center, Vagelos College of Physicians and Surgeons, Columbia University Irving Medical Center, 1130 St Nicholas Ave, New York, NY 10032, USA; 3Department of Neurological Surgery, Oregon Health & Sciences University, 3181 SW Sam Jackson Pkwy, Portland, OR 97239, USA; 4Department of Neurosurgery, Vagelos College of Physicians and Surgeons, Columbia University Irving Medical Center, 710 West 168th Street, New York, NY 10032, USA

**Keywords:** solitary fibrous tumor, hemangiopericytoma, radiotherapy, risk stratification

## Abstract

**Simple Summary:**

A solitary fibrous tumor (SFT)/hemangiopericytoma (HPC) of the central nervous system (CNS) represents a rare meningeal tumor with the propensity to recur almost invariably and to metastasize extracranially. Given the rarity of the disease, there are no prospective trials by which to guide its management, and indications for radiotherapy are unclear. The NRG Oncology and European Organization for Research and Treatment of Cancer (EORTC) cooperative groups recently completed the first prospective trials to evaluate risk-adapted radiotherapeutic strategies for meningiomas, based on tumor grade and extent of resection. Using a similar approach, we created three risk categories using two large national US datasets. Our risk categories were highly prognostic of overall and cause-specific survival. Furthermore, our risk categories predicted the survival benefit associated with radiotherapy, which was limited to the high-risk group and, potentially, the intermediate-risk group. Our data suggest that a risk-adapted approach may be employed for the management of SFT/HPC of the CNS. These risk categories may be used in future retrospective and/or prospective studies.

**Abstract:**

Introduction: Solitary fibrous tumor/hemangiopericytoma (SFT/HPC) of the central nervous system (CNS) is a rare meningeal tumor. Given the absence of prospective or randomized data, there are no standard indications for radiotherapy. Recently, the NRG Oncology and EORTC cooperative groups successfully accrued and completed the first prospective trials evaluating risk-adapted adjuvant radiotherapy strategies for meningiomas. Using a similar framework, we sought to develop prognostic risk categories that may predict the survival benefit associated with radiotherapy, using two large national datasets. Methods: We queried the National Cancer Database (NCDB) and the Surveillance, Epidemiology, and End Results (SEER) databases for all newly diagnosed cases of SFT/HPC within the CNS. Risk categories were created, as follows: low risk—grade 1, with any extent of resection (EOR) and grade 2, with gross–total resection; intermediate risk—grade 2, with biopsy/subtotal resection; high risk—grade 3 with any EOR. The Kaplan–Meier method and Cox proportional hazards regressions were used to determine the association of risk categories with overall and cause-specific survival. We then determined the association of radiotherapy with overall survival in the NCDB, stratified by risk group. Results: We identified 866 and 683 patients from the NCDB and SEER databases who were evaluated, respectively. In the NCDB, the 75% survival times for low- (*n* = 312), intermediate- (*n* = 239), and high-risk (*n* = 315) patients were not reached, 86 months (HR 1.60 (95% CI 1.01–2.55)), and 55 months (HR 2.56 (95% CI 1.68–3.89)), respectively. Our risk categories were validated for overall and cause-specific survival in the SEER dataset. Radiotherapy was associated with improved survival in the high- (HR 0.46 (0.29–0.74)) and intermediate-risk groups (HR 0.52 (0.27–0.99)) but not in the low-risk group (HR 1.26 (0.60–2.65)). The association of radiotherapy with overall survival remained significant in the multivariable analysis for the high-risk group (HR 0.55 (0.34–0.89)) but not for the intermediate-risk group (HR 0.74 (0.38–1.47)). Similar results were observed in a time-dependent landmark sensitivity analysis. Conclusion: Risk stratification based on grade and EOR is prognostic of overall and cause-specific survival for SFT/HPCs of the CNS and performs better than any individual clinical factor. These risk categories appear to predict the survival benefit from radiotherapy, which is limited to the high-risk group and, potentially, the intermediate-risk group. These data may serve as the basis for a prospective study evaluating the management of meningeal SFT/HPCs.

## 1. Introduction

Solitary fibrous tumor (SFT)/hemangiopericytoma (HPC) of the central nervous system (CNS) is a rare meningeal tumor with an incidence rate of 3.8 cases per 10,000,000 persons per year in the US, which is rising [1,2]. The incidence rate approached 6 persons per 10,000,000 persons per year in 2013, or approximately 230 cases diagnosed annually. In 2016, the World Health Organization (WHO) created a combined designation of SFT/HPC, recognizing that the two tumors share the *NAB2*/*STAT6* fusion and, therefore, likely represent tumors with a common genetic etiology along a spectrum of possible clinical behaviors [3]. Unlike meningiomas and low-grade solitary fibrous tumors [4,5,6], hemangiopericytomas recur almost invariably [7,8,9,10,11,12,13,14,15] and have a high propensity for extracranial metastasis [15]. The most recent WHO update in November 2021 (CNS-5) removed the term “hemangiopericytoma” so that the tumor name would conform fully with soft-tissue pathology nomenclature [16]. 

Optimal management of CNS SFT/HPC includes maximal safe resection, with or without adjuvant radiotherapy [1]. Because there are no randomized controlled trials or prospective studies, indications for adjuvant radiotherapy remain unclear and are institution-dependent. Adjuvant radiotherapy is administered for approximately 53% of HPCs classified as grades 2–3 in the US [1,17]. Retrospective series, population-based studies, and meta-analyses have yielded mixed results regarding the survival benefit of radiotherapy, likely due to selection bias and confounding clinical factors [8,9,13,14,15,18,19,20,21,22,23,24,25,26,27,28]. Adjuvant radiotherapy is more likely to benefit patients with higher-grade tumors or a lesser extent of resection (EOR) [1]; however, there is no consensus on the absolute indications. Retrospective real-world datasets are unlikely to simulate clinical trial outcomes [29]. Prospective studies to help guide management are, therefore, vitally needed.

NRG Oncology (formerly known as the Radiation Therapy Oncology Group (RTOG)) and the European Organization for Research and Treatment of Cancer (EORTC) have now both successfully enrolled and completed the first prospective, non-randomized phase-II trials evaluating adjuvant radiotherapeutic strategies for meningiomas [30,31,32,33,34]. These trials have been used to create risk-adapted standardized treatments and are also the basis for ongoing randomized clinical trials, thus laying the groundwork for evidence-based management of meningiomas.

No similar risk-adapted strategies have been developed for SFT/HPC. Here, we propose a risk-stratification schema for SFT/HPC, which may be considered as a foundation for future prospective or retrospective studies, with the intention of developing more standardized treatment paradigms. Similar to the RTOG and EORTC trials, we formulated prognostic risk groups based on tumor grade and EOR. We hypothesized that risk stratification could model prognosis better than any one individual clinical feature and thereby predict the survival benefit from radiotherapy. As a result, we analyzed risk categories and treatment-related outcomes reported in two large national databases. 

## 2. Materials and Methods

### 2.1. Data Sources

The National Cancer Database (NCDB) is a retrospective nationwide dataset sponsored by the American College of Surgeons and the American Cancer Society, constituting 70% of invasive cancer cases diagnosed in the United States. Data were collected at over 1500 Commission on Cancer–accredited hospitals between 2004 and 2018 [35]. This database has been validated for several variables [36,37,38,39].

The Surveillance, Epidemiology, and End Results (SEER) program is the National Cancer Institute’s (NCI) authoritative source for data on cancer incidence and survival [40]. It is considered the gold standard for cancer data collection internationally [36]. The SEER 18 database is populated with data from national cancer registries in 13 states, covering approximately 27.8% of the United States population [40]. The Commission on Cancer of the American College of Surgeons requires the participating cancer registries to collect information on malignancies that are diagnosed and/or treated at the hospital. Vital status is updated annually and the database routinely undergoes quality-control checks. Our methodology was conducted as described previously [41,42,43,44,45,46,47].

### 2.2. Patient Selection and Coding

We queried the NCDB (2018 submission) to identify cases of SFT (International Classification of Diseases (ICD)-O-3 code 8815) and HPC (ICD-O-3 code 9150) within the CNS (ICD-O-3 codes C70.1–C72.9) diagnosed between 1 January 2004 and 31 December 2016. The last possible date of follow-up for all cases was 31 December 2018. The following variables were collected and coded: age at diagnosis, sex, race, Charlson–Deyo score, primary site, tumor size, ICD-O-3 histology, ICD-0-3 behavior, collaborative staging (CS) site-specific factor 1 (WHO grade), surgery at the primary site, and radiation therapy. 

Grades were determined using all the available information from ICD-O-3 histology, ICD-0-3 behavior, and CS site-specific factor 1 (WHO grade), to keep them consistent with the WHO 2016 grading criteria. All primary tumors reported to US cancer registries contain both a 4-digit ICD-0-3 histology code and a 5th digit for ICD-0-3 behavior. Behavior coding is based on histological morphology and indicates the likely behavior of the tumor in terms of its potential to invade the surrounding tissue, based on the behavior that most pathologists believe is usual for that tumor type. ICD-0-3 behavior coding can be changed at the discretion of the coding pathologist. Tumors are classified as benign, borderline malignant, or malignant. Tumors are coded as borderline malignant based on a pathologist’s observations that the tumor has “low, borderline, or uncertain malignant potential”. Based on the WHO 2016 grading criteria, SFTs were coded as grade 1 and HPCs as grade 2, unless the tumors displayed malignant behavior, in which case they were coded as grade 3. This was compared with the WHO grade when it was available, and the findings were generally concordant. In cases where the histology/behavior codes were discordant with the WHO grade, the WHO grade was used. Information on molecular analysis, including STAT6 immunostaining and/or *NAB2*-*STAT6*, was not available. 

The extent of resection was based on definitions in the American College of Surgeons Commission on Cancer’s Facility Oncology Registry Data System (FORDS) manual [48]. Primary site surgeries in US cancer registries are defined as “cancer-directed” if the goal of treatment is to modify, control, remove, or destroy cancer tissue. Incisional biopsies are not considered to be cancer-directed surgeries. Most patients that did not undergo cancer-directed surgeries had received histological confirmation of disease and were, therefore, assumed to have undergone biopsy. EOR was coded as a biopsy/STR or GTR, based on surgery with the primary site variable: “no surgery” (code 00 (no surgery of the primary site)), “subtotal resection” (STR) (codes 10 (tumor destruction, not otherwise specified), 21 (STR), 20 (local excision or excisional biopsy), 22 (resection of the tumor in the spinal cord or nerve), 40 (partial resection of the lobe of the brain when surgery cannot be coded as 20–30)), and “gross-total resection” (GTR) codes (30 (radical, total, gross resection of the tumor) and 55 (GTR of a lobe of the brain)), as is consistent with prior studies [1,49,50,51,52,53]. Because surgical coding in cancer registries is based on the anatomical extent of the resection and not on the residual tumor, we performed multiple sensitivity analyses using different EOR coding schemas. 

We combined the grade and EOR variables and then further grouped those cohorts of patients with similar overall survival prognoses. Risk categories were created as follows: low risk—grade 1 with any EOR, grade 2 with GTR; intermediate risk—grade 2 with biopsy/STR; high risk—grade 3 with any EOR. 

Patients were excluded if follow-up time was less than two months as these patients either did live long enough to undergo adjuvant treatment or see an effect of management. We also excluded patients with metastatic disease or that could not be defined by our risk-stratification schema.

We also queried the SEER 17 database (November 2021 submission (2000–2019)) [54] for newly diagnosed cases of SFT/HPC that were diagnosed between 1 January 2000 and 31 December 2019, with follow-ups through December 2020. The following variables were collected and coded: age at diagnosis, sex, race, ICD-O-3 histology, ICD-0-3 behavior, primary site, surgery at the primary site, and collaborative staging (CS) site-specific factor 1 (WHO grade). Uniform coding across all years of analysis was not available for other potentially prognostic variables, such as size. Cases diagnosed at autopsy, or that could have 0 days of follow-up, and cases with less than 1 month of follow-up were excluded, as were cases that could not be defined by our risk-stratification classification. 

### 2.3. Statistical Analysis

Median survival times were determined using the Kaplan–Meier method, and significance was determined using the log-rank test. The 75th percentile survival time was used as a surrogate marker of survival when the median survival time was not reached [49,50,55]. Univariable and multivariable analyses of both overall survival (OS) and cause-specific survival (CSS) were conducted using the Cox proportional hazards ratios model, with logistic regressions. The 95% confidence intervals were expressed next to the corresponding hazard ratios (HR). Tests with two-tailed *p*-values < 0.05 were considered to be statistically significant. Demographic and clinical features that were significantly associated with survival were included in the multivariable analyses. Statistical analyses were conducted using SEER*Stat, version 8.3.9 (National Cancer Institute, Bethesda, MD, USA) and RStudio version 1.4.1106 (R-Project for Statistical Computing, Boston, MA, USA) software.

## 3. Results

### 3.1. Patient Selection and Clinical/Demographic Characteristics 

We identified 1,578 patients in the NCDB who were newly diagnosed with SFT or HPC. After excluding those patients with metastatic disease (*n* = 32), less than two months of follow-up (*n* = 163), or unknown EOR (*n* = 479), there were 866 patients available for analysis. The median follow-up time for all cases was 44 months, with 149 deaths. Demographic and clinical characteristics of the patient population are displayed in Table 1. 

In SEER, there were 715 cases of SFT/HPC. After excluding those patients diagnosed at autopsy, who could be considered to have 0 days of follow-up or less than 1 month of follow-up (*n* = 19), or an unknown extent of resection (*n* = 13) were excluded, 683 cases were available for analysis. The median follow-up time was 66 months. There were 197 recorded deaths, 62 of which were attributed to SFT/HPC (Appendix A).

### 3.2. Development of Risk Stratification Model

In NCDB, the median survival time for all patients was not reached, with a 75% survival time of 86 months. The 75% survival times for tumors of grades 1, 2, and 3 were 92, 89, and 55 months, respectively (*p* = 0.001) (see Appendix A). Compared with grade 1, grade 3 (HR 2.53 (95% CI 1.37–4.66), *p* = 0.003) but not grade 2 (HR 1.34 (95% CI 0.72–2.48), *p* = 0.36) disease was associated with poorer survival rates. GTR was associated with an improved OS rate compared with biopsy/STR patients (75% survival time of 99 vs. 68 months, HR 0.59 (95% CI 0.42–0.84), *p* = 0.003).

We combined the grades and EOR to create the following risk categories: low risk—grade 1 with any EOR and grade 2 with GTR; intermediate risk—grade 2 with biopsy/STR; high risk—grade 3 with any EOR. These risk categories improved the prognostic value compared with any single risk factor. The 75% survival times for low-, intermediate-, and high-risk tumors were calculated as not reached, 86 months, and 55 months, respectively (*p* < 0.001, Figure 1A). Compared with low-risk disease, intermediate-risk, and high-risk disease were associated with poorer OS on univariable (HR 1.60 (95% CI 1.01–2.55), *p* = 0.05 and HR 2.56 (95% CI 1.68–3.89), *p* < 0.001, respectively (see Table 2 and Appendix A)), and multivariable analysis (HR 1.52 (95% CI 0.95–2.41), *p* = 0.08 and HR 2.38 (95% CI 1.56–3.63), *p* < 0.001, respectively). 

We sought to validate our risk-stratification model in SEER. Although there may be an overlap of patients in SEER and NCDB, these databases use fundamentally distinct mechanisms to collect patient data; they undergo different quality-control processes and contain different variables. In the SEER dataset, risk stratification also improved the prognostic modeling over any single risk factor (Figure 1B and Appendix A, *p* < 0.001). The 75% survival times for low-, intermediate-, and high-risk patients were 119, 88 (HR 1.90 (95% CI 1.25–2.90), *p* = 0.003), and 51 months (HR 2.76 (95% CI 1.86–4.08), *p* < 0.001), respectively (Table 2 and Appendix A). When evaluating CSS, risk stratification was also associated with improved prognostication, compared with individual clinical factors (Figure 1C). There were 0, 2 (1.3%), and 33 (20%) cause-specific deaths in the low-, intermediate-, and high-risk groups, respectively, with the corresponding 75% survival times not reached, not reached, and 111 months (*p* < 0.001), respectively. Given that there were no events in the low-risk group, the corresponding HRs could not be calculated. 

### 3.3. Risk Stratification Predicts Benefit of Radiotherapy

Of the 859 patients with known radiotherapy status (99.2%), 425 (49%) received radiotherapy. Across all patients in the NCDB dataset, radiotherapy was not associated with improved OS (75% survival times of 89 vs. 73 months, HR 0.84 (0.61–1.17), *p* = 0.30, see Appendix A). However, when stratifying according to risk group, radiotherapy was associated with an improved OS in the high-risk (75% survival time 78 vs. 33 months, HR 0.46 (0.29–0.74), *p* = 0.001) and intermediate-risk groups (89 vs. 66 months, HR 0.52 (0.27–0.99), *p* = 0.05), but not in the low-risk group (not reached vs. not reached, HR 1.26 (0.60–2.65), *p* = 0.55, Figure 2, Table 3, and Appendix A). With the multivariable analysis, radiotherapy remained associated with an improved OS in the high-risk group (HR 0.59 (0.36–0.95), *p* = 0.03) but not in the intermediate-risk group (HR 0.74 (0.38–1.47), *p* = 0.39). 

### 3.4. Sensitivity Analyses

ICD-O-3 histology and behavior were available for 100% of cases. Additional information on WHO grades was available for 558 (64.4%) patients in the NCDB. The grade of the tumor was modified for 126 (22.5%) cases when the WHO grade was included. We performed a sensitivity analysis, excluding those patients with missing data. Similar results were observed when we only included those patients with all histological data points (Appendix A). 

Because patients who died very early would not have had the opportunity to undergo radiotherapy, our outcome of interest may be affected by the immortal time bias [56]. To address the immortal time bias, we performed a sequential landmark time analysis. For patients who received radiotherapy, the time from diagnosis to the initiation of radiotherapy was available for 414 (97.4%) patients. The median time from diagnosis to the initiation of radiotherapy was 60 days, suggesting that our initial exclusion period of patients with less than 2 months of follow-up was appropriate. We performed additional landmark analyses for those patients with at least 3 and 6 months of follow-up. Similar results were observed in the high-risk group, with HRs of 0.51 (0.31–0.84, *p* = 0.008, Appendix A) and 0.54 (0.32–0.90 *p* = 0.02, Appendix A), respectively. In the intermediate-risk group, radiotherapy was no longer associated with a statistically significant improvement in OS in the 3-month (HR 0.57 (HR 0.29–1.11), *p* = 0.10, Appendix A) and 6-month analyses (HR 0.65 (0.32–1.31), *p* = 0.23, Appendix A). 

## 4. Discussion

In this study, we develop and validate a risk-stratification schema for SFT/HPCs of the CNS according to the WHO 2016 histological guidelines, stratified by grade and EOR. Our risk categories were prognostic of OS and CSS and predicted outcomes better than any single clinical factor. Furthermore, our risk categories stratified patients to determine the survival benefit associated with radiotherapy. These risk categories may be used in future prospective trials or retrospective studies that evaluate the survival benefit of adjuvant radiotherapy.

The OS advantage observed with radiotherapy in the univariable analysis was limited to those patients with intermediate- or high-risk disease. Low-risk patients had a comparatively favorable prognosis and did not seem to experience a survival benefit from radiotherapy. Furthermore, at a median follow-up time of 80 months in the SEER dataset, there were no cause-specific deaths in the low-risk group. This suggests that radiotherapy can potentially be deferred in low-risk patients without affecting survival. 

For high-risk tumors, the prognosis was poor, with most cause-specific deaths occurring in this group. Radiotherapy was associated with a reduction in mortality by over 50%, suggesting that radiotherapy is essential for disease management. This benefit remained robust on multivariable analysis and in multiple landmark sensitivity analyses. Given the poor prognosis of the disease, regardless of treatment, clinical trials that access treatment-escalation in this group beyond adjuvant radiotherapy may be appropriate. 

In the intermediate-risk group, radiotherapy was associated with improved OS in the univariable analysis. However, it was no longer associated with improved OS in the multivariable analysis when including patient age, suggesting that the survival advantage may be confounded by patient selection. Additionally, the association was no longer statistically significant in our time-dependent landmark sensitivity analysis, which further supports the notion that patient selection is at least partially driving the observed effect. There were few cause-specific deaths in the intermediate-risk group in the SEER dataset at a median follow-up time of 74 months. Still, we cannot rule out the possibility that radiotherapy improves survival at later time points after 10 years. Even if radiotherapy does not improve OS, a progression-free survival benefit cannot be ruled out. Prolonging the time to progression may be associated with decreased morbidity and should be weighed against the potential toxicity from adjuvant radiotherapy. Toxicity datahave been reported in the RTOG and EORTC studies at dose levels of 54 and 60 Gy [30,31,34]. In the absence of randomized data, adjuvant radiotherapy should be considered at the clinical discretion of the treating provider, after a discussion of the risks and benefits. 

Our study was inspired by the RTOG and EORTC trials, which successfully enrolled and completed prospective studies on risk-adapted radiotherapeutic strategies for meningiomas. These trials created established standard protocols for the treatment of meningioma and also led to two randomized phase-III trials in the US and Europe. The ongoing NRG-BN003 and EORTC/ROAM trials will evaluate the role of adjuvant radiotherapy in grade-II meningiomas that undergo GTR. RTOG 0539 also demonstrated the feasibility of recruiting a high-risk group of patients with grade-III meningiomas, which are relatively rare [31]. Using a similar framework, we applied risk-stratification classes to SFT/HPC, based on prognostic groupings. Due to the rarity of SFT/HPC, a prospective study is unlikely. However, with an estimated 230 cases of SFT/HPC per year, versus 320 cases of malignant meningioma, a prospective trial may be feasible [1,2,57]. The last available estimate of the incidence rate is from 2013, and the incidence rate may have risen since then. Unlike extracranial SFT/HPC, the incidence rate of CNS SFT/HPC is slightly higher in Asian/Pacific Islanders [1,2]. Large series of CNS SFT/HPCs have been published from Asian countries and recruitment for trials may be more feasible in Asia. 

Risk categories in our study were developed based on the overall survival prognosis and not on progression-free survival and varied from the RTOG 0539 study as follows: grade 2 SFT/HPC tumors with GTR were categorized as low-risk in our study, whereas grade 2 meningiomas that underwent GTR were considered intermediate-risk in the RTOG study; grade 3 SFT/HPCs with GTR were considered high-risk in the RTOG study, whereas they were classified as intermediate risk in the current study. Additionally, we only analyzed newly diagnosed tumors, whereas the RTOG 0539 study included recurrent tumors as well. 

The advantages of our study include the use of two large national datasets. NCDB covers 70% of the US population and contains detailed treatment information, whereas SEER covers 28% of the US and is representative of the population. It also has cause-specific death information. Because we analyzed patients from 2004–2016 in the NCDB and 2000–2019 in SEER, we expected a considerable overlap of patients. These analyses were intended to be complementary, as data collection and quality control differ and because different variables are available.

The limitations of our study include retrospective analysis. Given the rarity of the tumor in question, there have been no prospective studies and and future prospective studiesare unlikely. A central histological review, including molecular analysis, was not possible. Although the EOR variable in the NCDB has been validated via data submitted from an academic center, the accuracy of EOR coding from nationwide samples is unknown [37]. Our analysis was corroborated in the SEER dataset, which may have better quality control procedures and less missing data. Radiotherapy may be under-coded in national datasets, which would bias our data toward the null hypothesis [1,17]. In the absence of prospective trials, large retrospective multi-institutional cohorts would be useful to validate our findings.

Our risk categories are pragmatic and may be applied in clinical scenarios when considering overall or cause-specific mortality for an individual patient. We advise that decisions for adjuvant treatment should be discussed within a multidisciplinary tumor board. 

## 5. Conclusions

SFT/HPC of the CNS is a rare meningeal tumor, with no current consensus on the standard of care for adjuvant management. In this study, we develop and validate a risk-stratification schema based on the tumor grade and EOR, which is similar to risk classes developed for the RTOG 0539 and EORTC 22042-26042 trials. Our risk categories were prognostic of OS and CSS and outperformed the prognostic capability of any individual risk factor. Furthermore, our risk groups were predictive of survival benefits from radiotherapy. Radiotherapy was associated with an improved OS in the intermediate- and high-risk groups but not in the low-risk group. There were no cause-specific deaths in the low-risk group, suggesting that radiotherapy can be deferred without affecting survival. The OS benefit was not statistically significant in themultivariable analysis or in our sensitivity analyses in the intermediate-risk group, suggesting that the survival benefit may be, at least partially, driven by patient selection. Still, radiotherapy may be associated with a progression-free survival benefit and this may translate into an OS benefit at later follow-up times. In the high-risk group, radiotherapy was associated with reduced mortality, suggesting that it is essential for the management of grade 3 tumors. Prognosis is poor for grade 3 tumors, and investigation of additional therapy-escalation may be warranted. 

These risk categories may be used as the basis for a prospective trial. Although a prospective study is unlikely, it may be feasible, given the rising incidence of SFT/HPC, the proof of feasibility already having been established when studying malignant meningioma in the RTOG 0539 trial. In the absence of prospective data, validation of our risk categories through a multi-institutional retrospective series would help in developing evidence-based management strategies for this rare tumor.

## Figures and Tables

**Figure 1 cancers-15-00876-f001:**
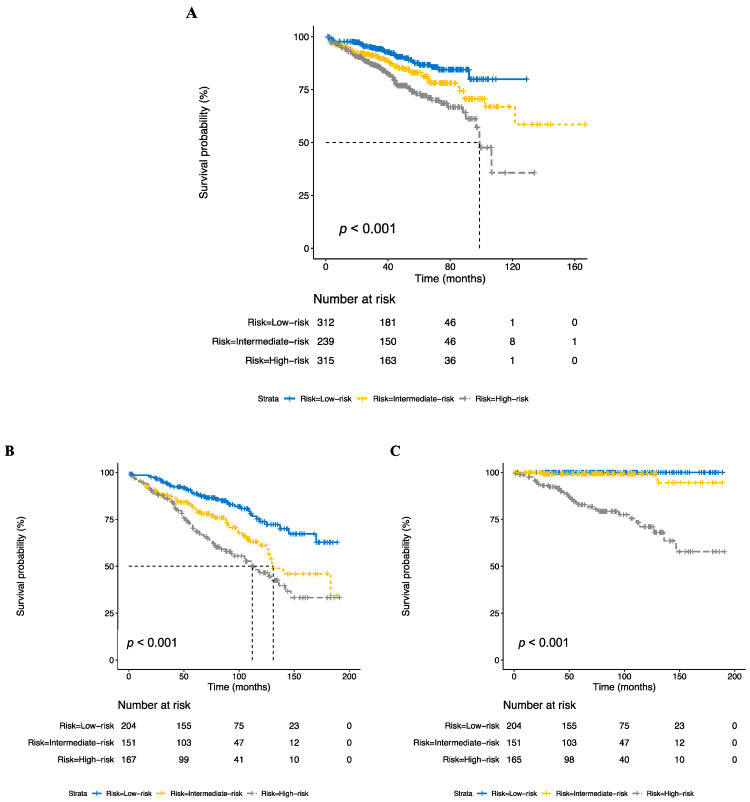
Kaplan–Meier curves representing overall survival in the NCDB (**A**) and overall (**B**) and cause-specific survival (**C**) in the SEER database based on risk groups.

**Figure 2 cancers-15-00876-f002:**
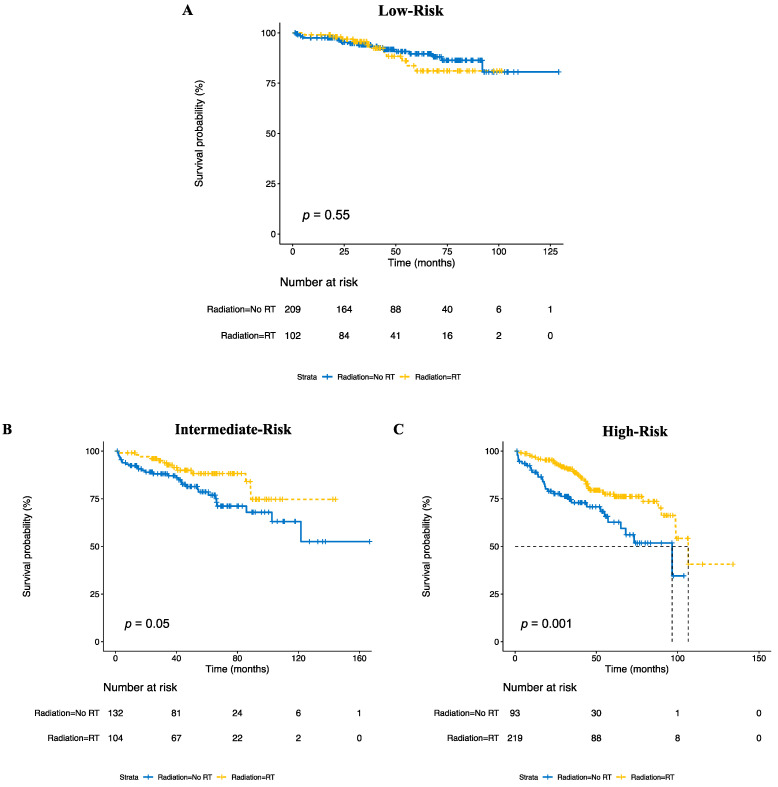
Kaplan–Meier curves for overall survival in the NCDB for low-risk (**A**), intermediate-risk (**B**), and high-risk (**C**) groups, based on the receipt of radiotherapy (RT).

**Table 1 cancers-15-00876-t001:** Demographic and clinical characteristics of the NCDB cohort.

Characteristic	Low-Risk, *N* = 312 ^1^	Intermediate-Risk, *N* = 239 ^1^	High-Risk, *N* = 315 ^1^
Age	54 (43, 65)	55 (43, 66)	54 (42, 66)
Sex			
Male	149 (48%)	112 (47%)	156 (50%)
Female	163 (52%)	127 (53%)	159 (50%)
Race			
White	252 (81%)	200 (84%)	267 (85%)
Black	25 (8.0%)	28 (12%)	26 (8.3%)
Other/Unknown	14 (4.5%)	4 (1.7%)	6 (1.9%)
Asian/Pacific Islander	20 (6.4%)	7 (2.9%)	16 (5.1%)
Unknown	1	0	0
Charlson–Deyo Comorbidity Index			
0	252 (81%)	187 (78%)	236 (75%)
1	46 (15%)	35 (15%)	47 (15%)
2 or more	14 (4.5%)	17 (7.1%)	32 (10%)
Site			
Brain	241 (77%)	168 (70%)	269 (85%)
Spinal/Other CNS	71 (23%)	71 (30%)	46 (15%)
Histology			
SFT	115 (37%)	0 (0%)	22 (7.0%)
HPC	197 (63%)	239 (100%)	293 (93%)
Grade			
G1	115 (37%)	0 (0%)	0 (0%)
G2	197 (63%)	239 (100%)	0 (0%)
G3	0 (0%)	0 (0%)	315 (100%)
Tumor Size			
5cm or less	128 (41%)	103 (43%)	118 (37%)
Greater than 5cm	110 (35%)	51 (21%)	91 (29%)
Unknown	74 (24%)	85 (36%)	106 (34%)
EOR			
No surgery/STR	72 (23%)	239 (100%)	163 (52%)
GTR	240 (77%)	0 (0%)	152 (48%)
Radiation			
No radiotherapy	209 (67%)	132 (56%)	93 (30%)
Radiotherapy	102 (33%)	104 (44%)	219 (70%)
Unknown	1	3	3
Follow-up Time	45 (29, 69)	49 (30, 74)	41 (26, 61)
Vital Status			
0	281 (90%)	195 (82%)	241 (77%)
1	31 (9.9%)	44 (18%)	74 (23%)

^1^ Median (IQR); *n* (%).

**Table 2 cancers-15-00876-t002:** Summary of the univariable and multivariable analyses of the risk groups in NCDB and SEER.

	Univariable			Multivariable		
Dataset/Characteristic	HR	95% CI	*p*-Value	HR	95% CI	*p*-Value
NCDB ^1^						
Low risk	-	-		-	-	
Intermediate risk	1.60	1.01, 2.55	**0.045**	1.52	0.95, 2.41	0.079
High risk	2.56	1.68, 3.89	**<0.001**	2.38	1.56, 3.63	**<0.001**
SEER ^2^						
Low risk	-	-		-	-	
Intermediate risk	1.90	1.25, 2.90	**0.003**	1.94	1.27, 2.95	**0.002**
High risk	2.76	1.86, 4.08	**<0.001**	2.62	1.76, 3.92	**<0.001**

The table provides summary statistics from the univariable and multivariable analyses. Full univariable and multivariable analyses are included in the Appendix A. Variables that were significant in the univariable analysis were included in the multivariable analyses. Bold values are statistically significant. ^1^ Variables included in the analysis were age, sex, race, the Charlson–Deyo comorbidity index, tumor size, anatomical site, risk group, and radiotherapy. ^2^ Variables included in the analysis were age, sex, race, anatomical site, and risk group.

**Table 3 cancers-15-00876-t003:** Summary of the association between radiotherapy and overall survival in NCDB, according to risk group.

Variable	Univariable			Multivariable		
	HR of Radiotherapy	95% CI	*p*-Value	HR of Radiotherapy	95% CI	*p*-Value
Risk Group						
Low risk	1.26	0.60, 2.65	0.55	-	-	
Intermediate risk	0.52	0.27, 0.99	**0.048**	0.74	0.38, 1.47	0.39
High risk	0.46	0.29, 0.74	**0.001**	0.59	0.36, 0.95	**0.031**

HR—hazard ratio, CI—confidence interval. The table provides summary statistics on the association of radiotherapy, with overall survival from separate univariable and multivariable analyses, stratified by risk group. Full univariable and multivariable analyses are included in the Appendix A. Variables included in the analysis were age, sex, race, Charlson–Deyo comorbidity index, tumor size, anatomical site, the extent of resection, and radiotherapy. Variables that were significant in the univariable analysis were included in the multivariable analyses. Bold values are statistically significant.

## Data Availability

The SEER and NCDB datasets are available upon request from the NCI and ACS. Coding scripts are available from the authors upon request.

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
