# Peer review of "Risk Stratification for Management of Solitary Fibrous Tumor/Hemangiopericytoma of the Central Nervous System"

_cancers, 2023, doi:10.3390/cancers15030876_

Round 1
Reviewer 1 Report
As the authors highlight, solitary fibrous tumor/hemangiopericytoma (SFT/HPC) are rare meningeal tumors with no prospective or randomized data. The indication for radiotherapy is uncertain. The author’s developed prognostic risk categories to predict benefit from radiotherapy in two large national datasets, National Cancer Database (NCDB) and the Surveillance, Epidemiology, and End Results (SEER).
For all newly diagnosed cases of SFT/HPC within the CNS, the risk stratified SFT/HPC as: (1)
Low risk, defined as Grade 1 with any extent of resection (EOR) or Grade 2 with gross-total resection; (2) Intermediate Risk, defined as Grade 2 with biopsy/subtotal resection; and (3) High risk, defined as Grade 3 with any EOR based on 866 NCDB and 683 SEER database patients.
NCDB risk categories were validated for overall and cause-specific survival in the SEER dataset. Radiotherapy was associated with improved survival in the high- (HR 0.46 [0.29-0.74]) and intermediate-risk groups (HR 0.52 [0.27-0.99]), but not the low-risk group (HR 1.26 [0.60-2.65]). This association remained significant on multivariable analysis only in the high-risk group (HR 0.55 [0.34-0.89]).
Risk-stratification based on grade and EOR was prognostic for OS and cause-specific survival, more so than any individual clinical factor. These risk categories also predicted survival benefit from radiotherapy. These data may impact trial design for the rare meningeal SFT/HPCs.
Author Response
We thank Reviewer #1 for the accurate and succinct description of our manuscript.
Reviewer 2 Report
In this manuscript, Kinslow et al. developed a risk-stratification schema including three prognostic risk categories (low risk, intermediate risk and high risk) to predict benefit in patients after radiotherapy for Solitary fibrous tumor (SFT)/hemangiopericytoma (HPC) of the central nervous 17 system (CNS). More specifically, they showed their risk categories can significantly stratify patient survival after radiotherapy in the NCDB and SEER databases. Their strategy is superior to any other individual clinical factor. The study is interesting, but I however have some considerable concerns that need be comprehensively addressed:
1. Are there any other reported risk stratification strategies for SFT/HPC in the literature? Is this the first one for SFT/HPC?
2. The authors should discuss the rationale of combining the grade level with EOR in their risk categories.
3. What is the difference between the risk-adapted strategies used in meningiomas and this strategy for SFT/HPC developed by the authors?
4. How many patients are overlapped between NCDB and SEER datasets? It should be mentioned in the manuscript.
5. The authors claimed this novel risk stratification schema works better than any other individual clinical feature, what about combined clinical features? Additionally, it would be interesting to test whether any other clinical features could add extra value to this schema.
6. The authors may consider building up a machine learning predictor of radiotherapy response based on these risk factors and clinical features for SFT/HPC.
7. It seems Figure 2 is not mentioned in the main article.
8. How feasible is it to apply this strategy to clinics? Can it be extended to other cancer types?
Reviewer 3 Report
In this paper, Kinslow and collagues the development of prognostic risk categories that may predict benefit from radiotherapy in two large datasets including 1549 patients with a diagnosis of solitary fibrous tumor/hemangiopericytoma of the central nervous system. The risk categories proposed seem to predict survival benefit from radiotherapy, which is limited to the intermediate- and high-risk groups.
The paper is interesting and well-written, but, in my opinion, this data should be confirmed in other prospective series, because i acknowledge the limitations of this study related to retrospective study design and because the histological grading criteria are changed in the last 2021 WHO Classification of Tumors of the Central Nervous System, where the number of mitosis for 2 mm2 and the presence of necrosis are fundamental.
For this, I suggest to avoid “2021” in line 257, page 7 “the CNS according to the most recent WHO 2016/2021 histological guidelines”, because the authors seem to apply only WHO 2016 histological guidelines of the classification of tumors of the central nervous.
Reviewer 4 Report
Kinslow et al report a study on solitary fibrous tumors and indications for radiotherapy. SEER and NCDB were utilized. Outcomes were assessed with stratification by extent of resection and tumor grade. Approximately 1500 patients were included. Radiotherapy had a survival benefit in moderate and high risk cases on univariate analysis, but only high risk patients on multivariate analysis. Reduced survival time means were also associated with higher risk level of tumor, as expected. This is a nice summary paper and worthy of publication.
Did 100% of patients within these database have all of the required histological results/codes reported to optimally address the modern grading schema? It would be worthwhile to specify what percentage of patients were missing data that would optimally be available for diagnosis (i.e prior to 2016 and lacking some of the necessary adjuvant data points for ideal grading even if some data was available to provide a presumed grade) since inevitably this results in some level of data heterogeneity the further back one goes. If there are sufficient patients that were enrolled under the most modern criteria and with all optimal histological codes, it might be worth doing a univariate analysis of these just to demonstrate that the trends hold with more homogenous data.
Some clarification of the multivariate analysis is required. What type of “multiviariable analysis” was performed? Survival is often optimally assessed using logistic regression but I get the impression that this may have been done using linear regression. It is not clear if each table has a separate multivariate analysis or if they were all performed together but displayed separately. Optimally a single table with the full multivariate analysis including all demographic variables reaching criteria for inclusion would be performed. I do not see any analysis of the variables in table 1 as predictors of outcome. Most importantly, radiation therapy needs to be part of an outcome analysis when other variables such as high grade are also present in order to demonstrate that both high grade and radiation therapy are independent predictors of outcome and not simply not confounding variables.
The authors conclude that intermediate and high risk cases benefit from radiation, but this should be dialed back as the intermediate group was not independently associated with a benefit from radiation on multivariate analysis.
Reviewer 5 Report
There are some comments,
1. Regarding terminology, solitary fibrous tumors (SFT) and hemangiopericytomas cases were separately described in Table 1. Is there any difference between SFTs and hemangiopericytomas cases?
2. It would be better to describe whether STAT6 immunostaining or molecular analyses were performed for the diagnosis of SFT.
3. It would be better to evaluate according to the criteria of the 2021 CNS WHO grading system for the histologic grade of SFT.
4. Please modify the format of the references according to Author's guidelines.
